# Natural Polyphenol-Containing Gels against HSV-1 Infection: A Comparative Study

**DOI:** 10.3390/nano12020227

**Published:** 2022-01-11

**Authors:** Mariaconcetta Sicurella, Maddalena Sguizzato, Paolo Mariani, Alessia Pepe, Anna Baldisserotto, Raissa Buzzi, Nicolas Huang, Fanny Simelière, Sam Burholt, Peggy Marconi, Elisabetta Esposito

**Affiliations:** 1Department of Chemical, Pharmaceutical and Agricultural Sciences, University of Ferrara, I-44121 Ferrara, Italy; scrmcn@unife.it (M.S.); sgzmdl@unife.it (M.S.); 2Department of Life and Environmental Sciences, Università Politecnica delle Marche, I-60131 Ancona, Italy; p.mariani@staff.univpm.it (P.M.); a.pepe@pm.univpm.it (A.P.); 3Department of Life Sciences and Biotechnology, University of Ferrara, I-44121 Ferrara, Italy; anna.baldisserotto@unife.it (A.B.); raissa.buzzi@unife.it (R.B.); 4CNRS, Institut Galien Paris-Saclay, Université Paris-Saclay, 92296 Châtenay-Malabry, France; nicolas.huang@universite-paris-saclay.fr (N.H.); fanny.simeliere@universite-paris-saclay.fr (F.S.); 5Diamond Light Source Ltd., Harwell Science and Innovation Campus, Didcot OX11 0DE, UK; sam.burholt@diamond.ac.uk

**Keywords:** quercetin, mangiferin, polyphenols, HSV-1, localized drug action, infection control, in vitro diffusion, poloxamer, lecithin, gels

## Abstract

Herpes simplex virus type 1 infection commonly affects many people, causing perioral sores, as well as severe complications including encephalitis in immunocompromised patients. The main pharmacological approach involves synthetic antiviral drugs, among which acyclovir is the golden standard, often leading to resistant virus strains under long-term use. An alternative approach based on antiviral plant-derived compounds, such as quercetin and mangiferin, demonstrated an antiviral potential. In the present study, semisolid forms for cutaneous application of quercetin and mangiferin were designed and evaluated to treat HSV-1 infection. Phosphatidylcholine- and poloxamer-based gels were produced and characterized. Gel physical–chemical aspects were evaluated by rheological measurements and X-ray diffraction, evidencing the different thermoresponsive behaviors and supramolecular organizations of semisolid forms. Quercetin and mangiferin diffusion kinetics were compared in vitro by a Franz cell system, demonstrating the different gel efficacies to restrain the polyphenol diffusion. The capability of gels to control polyphenol antioxidant potential and stability was evaluated, indicating a higher stability and antioxidant activity in the case of quercetin loaded in poloxamer-based gel. Furthermore, a plaque reduction assay, conducted to compare the virucidal effect of quercetin and mangiferin loaded in gels against the HSV-1 KOS strain, demonstrated the suitability of poloxamer-based gel to prolong the polyphenol activity.

## 1. Introduction

In recent years, pharmaceutical research has increasingly rediscovered the possibility of using plant-derived substances for therapeutic purposes instead of synthetic molecules, based on their more favorable potential in terms of safety and efficacy [1]. In this regard, natural molecules can be an efficacious alternative in case of drug resistance, resulting, for instance, in the treatment of some viral infections, such as that induced by HSV-1, whose handling has become challenging [2,3,4]. Indeed, the prolonged treatment of this widespread and recurrent infection, usually based on acyclovir administration, can be a topic of concern, especially in the case of immunocompromised patients [5,6,7]. Immunosuppression can in fact lead to the development of antiviral drug resistance, resulting in persistent viremia or severe complications [8,9]. In this regard, natural polyphenols present in fruits, vegetables, nuts, tea, propolis and wine can be considered as promising alternatives to synthetic drugs [10]. Different studies have demonstrated that polyphenols display anti-infective properties, avoiding serious toxicity effects, a main feature required for antiviral agents [11,12,13]. Indeed, due to their antioxidant properties, polyphenols can be employed in the prevention or treatment of diseases correlated to oxidative stress [14]. For instance, quercetin (QT) and mangiferin (MG), as well as being powerful antioxidants and anti-inflammatories, have shown antiviral activity, such as in the treatment of HSV-1, being able to inhibit its replication [15,16,17,18]. Despite their therapeutic potential, both QT and MG are characterized by poor water solubility, limited skin permeation capacity and ease of oxidative degradation [19,20]. In this respect, special care should be devoted to designing a vehicle suitable for polyphenol solubilization and topical administration in the treatment of HSV-1. The painful lesions caused by HSV-1 affect mainly the lips, interfering with drinking and eating. An ideal dosage form for HSV-1 treatment should possess a suitable viscosity to be easily applied, while keeping adhesion on the lips to prolong the contact time between drug and lesion [21,22]. Furthermore, in order to improve polyphenol permeability and to reduce the frequency of application, providing longer therapeutic efficacy, specialized semisolid forms should be proposed over conventional ones. In this regard, in the present study different semisolid formulations composed of phosphatidylcholine (PC) and poloxamer 407 (P407) have been investigated. The low toxicity and biocompatibility of these surfactants make them markedly suitable for pharmaceutical applications [23]. PC is a natural zwitterionic surfactant with penetration enhancer properties, due to its chemical similarity with *stratum corneum* lipids [24,25]. Its capability to form different lyotropic crystalline phases in the presence of water (e.g., direct and inverted micelles or hexagonal, cubic and lamellar phases) makes PC suitable to form supramolecular structures for solubilization and controlled delivery of drugs [26]. Particularly, the addition of precise amounts of water to PC solution in biocompatible solvents results in the so-called organogels (OGs), isotropic delivery systems with a modulated viscosity, according to the water content. Indeed, PC under the addition of water self-organizes in reverse micelles that grow longitudinally, forming a three-dimensional network that thickens the oleaginous dispersing phase [27]. Different studies have demonstrated the huge OG transdermal potential [27,28,29]. On the other hand, hydrophilic semisolid forms can be prepared by P407, a block co-polymer belonging to the non-ionic surfactant category, being constituted by a polar central poly(oxyethylene) portion surrounded by two apolar poly(oxypropylene) tails [30]. P407 in water (20–30%, *v*/*v*) forms direct micelles that can solubilize drugs with different physico-chemical properties, undergoing transition to gel with a close micelle packing at a specific temperature, as a function of polymer concentration. This thermosensitive behavior confers to P407 gel (POL) a viscosity suitable for easy application on the skin at low temperature, while the contact with body temperature leads to a thickened form suitable to control drug release [31,32]. Moreover, a peculiar semisolid form can be obtained combining both OG and POL, resulting in poloxamer organogel (POG), characterized by a complex supramolecular structure, due to an entanglement of different lyotropic phases. The association of OG and POL components forms a biphasic system suitable for solubilization of both hydrophilic and lipophilic drugs and able to promote their permeation, due to an affinity with the bilayer lipid packing of the stratum corneum [33,34]. Recently, we have conducted a preformulative study to select the composition of smart semisolid forms suitable for MG application on skin and mucosae in the treatment of HSV-1 [35]. In the present investigation, OG, POL and POG were compared as delivery systems for QT and MG. Particularly, rheology studies shed light on gel thermosensitive behavior, while the inner structure of gels was evaluated by X-ray diffraction. The effect of drug loading in semisolid forms was investigated on their in vitro diffusion and capability to control chemical degradation. Finally, the biological efficacy of QT- or MG-loaded forms was evaluated, considering in vitro antioxidant capacity and the virucidal effect against the HSV-1 KOS strain.

## 2. Materials and Methods

### 2.1. Materials

Quercetin (3,3′,4′,5,6-Pentahydroxyflavone, 2-(3,4-Dihydroxyphenyl)-3,5,7-trihydroxy-4H-1-benzopyran-4-one, QT), mangiferin (1,3,6,7-Tetrahydroxyxanthone C2-β-D-glucoside, MG), isopropylpalmitate (IPP), the copolymer poly(ethylene glycol)-block-poly(propylene glycol)-block-poly(ethylene glycol) poloxamer 407 (P407) (PEO98-POP67-PEO98) and polytetrafluoroethylene (PTFE) membranes (2 cm diameter, pore size 0.2 μm) were purchased from Merck Life Science S.r.l. (Milan, Italy). The soybean lecithin (PC) (90% phosphatidylcholine) Epikuron 200 was a kind gift of Cargill (Minneapolis, MN, USA). Solvents were of HPLC grade, and all other chemicals were of analytical grade.

### 2.2. Preparation of Semisolid Forms

Semisolid forms were prepared as previously reported [35]. Briefly, to prepare lecithin organogel (OG), PC (200 mM) was solubilized in isopropylpalmitate (PC/IPP), and afterwards water (1.44% *v*/*v*) was added to the PC/IPP solution under stirring at 25 °C for 10 min. In the case of QT- and MG-containing organogel (OG-QT and OG-MG), the drugs were solubilized in PC/IPP solution before water addition. Poloxamer gel (POL) was prepared gradually by adding P407 (20% *w*/*w*) to cold water (5–10 °C), under magnetic stirring. The container was sealed and left in a refrigerator overnight at 4 °C. In the case of QT- and MG-containing gel (POL-QT and POL-MG) the drugs were added into the preformed P407 gel and solubilized under stirring at 25 °C. Poloxamer lecithin organogel (POG) was based on an aqueous phase constituted of POL and a lipid phase constituted of PC/IPP. To prepare POG, preformed POL was gradually added to PC/IPP under stirring, resulting in a 30:70 *v/v* ratio between POL and PC/IPP. In the case of QT- and MG-containing POG (POG-QT and POG-MG), the drugs were solubilized in POL before the addition into PC/IPP.

### 2.3. Rheological Measurements

Rheological measurements of OG, POL and POG were performed with an AR-G2 controlled-stress rotational rheometer (TA Instruments, New Castle, DE, USA) [34,35]. For POL and OG, the geometry used was a sand-blasted titanium aluminum cone–plate with a diameter of 40 mm, an angle of 1° and a truncation of 28 µm. For POG, instabilities were observed when using a cone–plate, so a sandblasted titanium plate–plate with a diameter of 40 mm was used for this gel. Both geometries were equipped with a solvent trap in order to prevent solvent evaporation during the experiments. The viscoelastic properties of the gels (elastic modulus G′ and viscous modulus G″) were assessed in oscillation mode. Oscillation frequency was set at 1 Hz and the deformations applied were all in the linear regime. Temperature ramps from 5 °C to 50 °C were obtained at a temperature rate of 1 °C/min and were controlled by a Peltier plate. Before starting the experiments, a 2 min conditioning time at 5 °C was applied. Measurements were performed at least thrice for each sample, to ensure reproducibility. As OG was too solid at 5 °C, OG samples were first placed on the lower plate at room temperature. The gap was then set, and the temperature was brought to 5 °C before the 2 min conditioning time at 5 °C.

### 2.4. X-ray Analyses

Small-angle and wide-angle X-ray scattering (SAXS and WAXS) experiments were performed using the Offline Xeuss 3.0 Diamond-Leeds SAXS facility (DL-SAXS) at Diamond Light Source (Harwell, UK). A gallium Excillum metaljet (9.4 keV), coupled with a movable beamstop-less Eiger 2 R 1M placed at the two distances of 100 mm and 2000 mm (for WAXS and SAXS, respectively) was used. The final investigated *Q*-range, (*Q* being the modulus of the scattering vector, defined as 4π sin θ/λ, where 2θ is the scattering angle) extended from 0.03 to 27.6 nm^−1^. The experiment exploited the mail-in service. OG, POL and POG samples were prepared in 2 mm polycarbonate capillaries, loaded into a capillary ladder, and analyzed at 25 °C. Each image had a 2 min collection time, with a 0.4 mm X-ray beam. Two-dimensional (2D) data were corrected for background, detector efficiency, a sample transmission and then radially averaged to derive I(Q) vs. Q curves.

### 2.5. Polyphenol Content of Semisolid Forms

QT and MG entrapment capacity (EC) was evaluated after 1 and 120 days from semisolid form preparation as previously reported [35]. Briefly, the forms were diluted with ethanol in a 1:10 *w*/*w* ratio, stirred for 30 min at 25 °C, and filtered by nylon syringe filters (0.22 μm pores). Polyphenol content was analyzed by HPLC as reported below, while the EC was calculated as follows:EC = P/T_P_ × 100,(1)
where P is the amount of polyphenol (QT or MG) determined by HPLC and T_P_ is the total amount of polyphenol weighed for the semisolid form production.

### 2.6. In Vitro Diffusion Experiments

In vitro polyphenol diffusion was investigated by Franz cells (Vetrotecnica, Padova, Italy) constituted of a lower receptor and an upper donor compartment separated by a PTFE membrane with exposed surface area of 0.78 cm^2^ (1 cm diameter orifice). The experiments were performed as previously reported [35,36], pouring 5 mL of receiving phase, constituted of ethanol/water (50:50, *v*/*v*), in the lower section. The membrane, previously hydrated with the receiving phase for 1 h, was mounted between the receptor section and the donor compartment, and afterwards approximately 1 g of P-containing semisolid form, or 1 mL of P ethanolic solution (30%, *v*/*v*), was placed in the donor compartment and sealed to avoid evaporation. In all samples, P was 0.5 mg/mL. During all the experiments, the receiving phase was stirred at 500 min^−1^ by a magnetic bar and thermostatted at 35 ± 2 °C [37]. At predetermined time intervals (0.5–7 h), 200 μL of receiving phase was collected, replaced with an equal volume and analyzed for polyphenol content by HPLC. The polyphenol concentrations were determined 6 times in independent experiments, calculating mean values ± standard deviations. Mean values were plotted as a function of time, obtaining the accumulation curves. The fluxes were extrapolated from the linear portion of the curves, considering the slopes of the regression line (angular coefficient). Finally, the diffusion coefficients were calculated according to Equation (2).
D = F/[P],(2)
where D is the diffusion coefficient, F is the flux and [P] is the polyphenol (QT or MG) concentration in the analyzed form, expressed in mg/mL.

### 2.7. HPLC Analysis

HPLC analyses were performed using Perkin Elmer Series 200 HPLC Systems equipped with a micro-pump, an auto sampler and a UV detector operating at 255 or 265 nm for QT or MG, respectively. A stainless-steel C-18 reverse-phase column (15 × 0.46 cm) packed with 5 μm particles (Hypersil BDS C18 Thermo Fisher Scientific S.p.A., Milan, Italy) was eluted at a flow rate of 1 mL/min with a mobile phase containing acetonitrile/water 40:60 *v*/*v*, in the case of QT, or methanol/water 60:40 *v*/*v*, in the case of MG. Before injection, the samples were appropriately diluted in the mobile phases.

### 2.8. Photochemiluminescence (PCL) Test

The PCL test measures the antioxidant activity of a sample against superoxide anion radicals with a Photochem^®^ apparatus (Analytik Jena, Leipzig, Germany) based on the method of Popov and Lewin [38]. The radicals are generated from luminol, a photosensitizing agent, which generates radicals upon exposure to UV light (Double Bore^®^ phosphor lamp, output 351 nm, 3 mW/cm^2^). Antioxidant capacity was measured at 25 °C using the manufacturer’s antioxidant capacity of liposoluble substance (ACL) kit. The kinetic light emission curve, which does not show a lag phase in ACL studies, was monitored for 180 s and expressed as μmol equivalent of Trolox (standard, TE) per gram of sample. The areas under the curves were calculated using the PCL soft control and analysis software. The samples were suitably diluted in methanol prior to analysis. The antioxidant assay was performed in triplicate and 10 μL of each sample diluted in HPLC grade methanol was sufficient to match the standard curve.

### 2.9. Antiviral Activity Study against HSV-1

#### 2.9.1. Cell Culture

Vero (African green monkey kidney) cells were cultivated in Eagle’s minimum essential medium (DMEM), supplemented with 5% fetal bovine serum (FBS), 100 mg/mL penicillin and 100 mg/mL streptomycin, incubated at 37 °C under 5% CO_2_ in an incubator. Cells were seeded at 5 × 10^5^ per well in a six-well plate, 24 h prior to plaque assay [35,39].

#### 2.9.2. Herpes Virus Stock Generation

Vero cells (2 × 10^7^) in 10–20 mL of cell culture medium were seeded into a 175 cm^2^ tissue culture flask and incubated overnight at 37 °C in a humidified 95% air–5% CO_2_ incubator. Vero cells were infected with herpes simplex virus strain KOS, at a multiplicity of infection (M.O.I) of 0.01. The cells were incubated for 1 h at 37 °C to allow adsorption of the virus to the cells. The flasks were rocked every 15 min to evenly distribute the inoculum. The virus inoculum was aspirated, and cell culture medium was added up to a final volume of 10–20 mL per flask. Infected cells were incubated at 37 °C in a humidified 95% air–5% CO_2_ incubator for 36–48 h, until complete cytopathic effect (CPE) was reached. Cells and supernatant were collected, cells were removed by low-speed centrifugation and the supernatant was centrifugated at 20,000 min^−1^ for 30 min. The obtained pellet was resuspended in 1 mL of medium, aliquoted and kept at −80 °C until use [35,39].

#### 2.9.3. Titration of Virus by Plaque Assay

The viral preparation was titrated on Vero cells. One day prior to titration, 6-well tissue culture plates with 0.5 × 10^6^ Vero cells per well were prepared. The virus was thawed on ice and sonicated for a few seconds prior to infection, to separate virus particles. A series of ten-fold dilutions (10^−2^–10^−10^) of the virus stock in 1 mL cell culture medium without serum in 1.5 mL Eppendorf tubes was prepared and added to each well containing cells. The cells were incubated for 1 h at 37 °C, to allow adsorption of the virus to the cells. After 1 h of infection, the viral inoculum was removed, and the monolayer was overlayed with 3 mL of 1% methylcellulose (Sigma-Aldrich, Milan, Italy). The plates were incubated for 3–4 days until well-defined plaques were visible. The methylcellulose medium was removed from the wells and stained for 10–20 min with 2 mL of crystal violet staining solution to fix the cells and the virus. The number of plaques was counted, the average for each dilution (n = 3) was determined and multiplied by 10 to the power of the dilution to obtain the number of plaque-forming units per mL (PFU/mL) [35,39].

#### 2.9.4. Antiviral Assay

The inhibition of virus replication (HSV-1 strain KOS 1 × 10^5^ PFU/mL) was measured by plaque assay, evaluating polyphenol-containing semisolid forms by virucidal assay.

#### 2.9.5. Virucidal Assay

HSV-1 KOS 1 × 10^5^ PFU/mL was incubated with SOL-QT, OG-QT, POL-QT, POG-QT, SOL-MG, OG-MG, POL-MG, POG-MG, OG, POL and POG, respectively, for 1 h and 6 h at 35 °C, before cell infection. In the drug-loaded formulations, QT and MG concentration was 50 µg/mL. Ten-fold dilutions of the mixture or virus alone were adsorbed to the cells for 1 h at 35 °C. After the adsorption time, the viral inoculum was removed and medium containing 1% methylcellulose was added. Plates were incubated for 3–4 days at 35 °C, washed and fixed with crystal violet, to determine the reduction in the number of plaques in the treated virus compared to the virus control [39,40].

#### 2.9.6. Statistical Analysis

All experiments were repeated 3 times and statistical values were expressed as the mean ± standard deviation (SD). For all data analyses, GraphPad Prism 9 software (GraphPad Software Inc., San Diego, CA, USA) was used. Values of *p* < 0.05 were considered statistically significant.

## 3. Results

### 3.1. Preparation and Characterization of Semisolid Forms

In order to design a vehicle suitable for topical administration of polyphenols in the treatment of HSV-1, in a previous study, semisolid forms containing MG were produced and characterized [35]. Special care was devoted to the choice of vehicles suitable for application and adhesion on skin and mucosae, possibly controlling drug release and permeability. Particularly, the surfactants PC and P407 were employed, due to their transdermal potential [27,33] and capability to form smart vehicles based on supramolecular assembly structures [35]. Three different formulations were evaluated: OG, an oleaginous gel based on PC and IPP, a hydrogel based on P407 (POL) and a poloxamer organogel (POG), that can be considered as a hybrid form of OG and POL.

In the present study, the performances of OG, POL and POG as delivery systems for QT or MG were compared. The production of semisolid forms was achieved by simple methods, with low energy consumption and devoid of the use of organic solvents. The vehicle compositions selected under the previous formulative study [35] are reported in Table 1.

In the case of OG, the addition of a precise amount of water to a PC/IPP solution quickly led to the formation of a viscous transparent gel. On the other hand, POL production was achieved by a previously reported cold method, resulting in a thermoreversible transparent gel, passing from the liquid to the semisolid state under heating [35]. Moreover, in the case of POG, the presence of both aqueous POL and oleaginous PC/IPP components conferred to the gel an opaque aspect and a thick consistency [33].

To gain information on the suitability of the semisolid forms for topical application, in a previous study, the viscosity values of OG, POL and POG were determined, measuring 0.11, 0.29 and 0.81 Pa·s at 1000 s^−1^, respectively [35]. In the present study, to further compare the rheological gel behavior, the viscoelastic properties were investigated by evaluating elastic (G′) and viscous (G′) moduli. Particularly, G′ and G″ profiles were plotted as a function of the temperature, determining the values of transition temperature T_sol–gel_. As shown in Figure 1, OG exhibited a complex profile with an overall elastic component, characterized by a first crossover in the steepest part of the curve, and another one in the more gradual slope, followed by a decrease in both G′ and G″.

This behavior suggests that the OG micellar network underwent a two-step breaking at two different temperatures, namely ≈11 °C and ≈17 °C (Table 2). At 11 °C, G′ and G″ dramatically decreased, while above 17 °C, G″ was higher than G’. Notably, above 32 °C, the G″ profile significantly overtook G′, suggesting for OG a fluid behavior rather than a solid one [27,28]. Conversely, the thermal behaviors of both POL and POG were quite different. Indeed, above the T_sol–gel_, G′ was predominant, indicating that on heating POL and POG became more elastic than viscous, mainly due to the thermosensitive behavior of P407, typically forming a structured gel [34]. Below the transition temperature, in the case of POL, G″ modulus predominated over G′, while in the case of POG, G′ and G″ profiles almost overlapped, probably because of the contribution given by the PC/IPP.

To shed light on the supramolecular structure of OG, POL and POG, SAXS and WAXS experiments were conducted at 25 °C, which represents the typical pharmaceutical storage temperature and is well above the T_sol–gel_ of the vehicles. SAXS profiles, reported in Figure 2, are particularly noisy, because of the low contrast and high dilution, however, differences are appreciable.

In the case of OG (Figure 2a), the absence of a peak suggests a PC disordered micellar or vesicular organization at 25 °C, in agreement with rheological findings evidencing a decrease in G′ and G″ profiles above the T_sol–gel_ (Figure 1a). Conversely, in the case of POL (Figure 2b), the occurrence of low-angle, narrow Bragg peaks indicates the presence of a well-organized 3D structure, indeed, above the T_sol–gel_, POL behaved like a gel (Figure 1b). As already observed [42], the solubilization of P407 in aqueous solution is due to dehydration of the hydrophobic POP blocks and hydration of the hydrophilic POE, resulting in micelle formation which can organize in an ordered 3D packing [43,44]. In particular, the peak indexing confirms the formation of a cubic micellar phase, with Pm3n symmetry (the so-called cubic Q223 phase) and a unit cell parameter of 28.2 nm (see Table 2) [45]. The structure of this phase consists of two types of disjointed micelles, packed in special positions of the cubic unit cell. Notably, according to the group symmetry, two quasi-spherical micelles are centered in the 223 space group positions *a* (edge and center of the cubic unit cell) while six disk-shaped micelles are centered in the positions *c* (two per face of the cubic unit cell) [45]. As already found, this phase has a type I micellar topology, e.g., the micelles are filled by the hydrophobic P407 blocks and are embedded in the water matrix. In the POG SAXS profile (Figure 2c), the band observed at about 0.03 Å^−1^ suggests the occurrence of a 3D packing of entangled micelles, even if the micellar network is less ordered than the one described for POL. Noticeably, the packing correlation distance is around 20 nm (see Table 2), as already observed in P407 gel supramolecular structures [41].

To further investigate the inner structure of the semisolid forms, WAXS analyses were performed. Figure 3 reports the scattering profiles of OG, POL and POG, showing bands probably characteristic of interchain distances of PC and POL. Namely, the OG profile (Figure 3a) is characterized by a wide peak centered at 1.5 Å^−1^, while the POL profile (Figure 3b) is characterized by a band at ca. 2 Å^−1^. Notably, in the case of POG (Figure 3c), constituted of both PC and P407, the profile clearly shows the superposition of the two bands, reflecting the contribution of both OG and POL components, in agreement with rheological findings. Repeat distances calculated from band positions are reported in Table 2.

To prepare OG-QT and OG-MG, the drugs were solubilized in the lipophilic phase constituted of PC/IPP before the addition of water, reaching 0.8 and 0.7 mg/mL for QT and MG, respectively. Conversely, the production of both the hydrophilic gels (POL-QT and POL-MG) and the biphasic systems (POG-QT and POG-MG) was achieved by drug solubilization in the preformed P407 aqueous solution. Notably, the micellar structure formed by the block co-polymer allowed solubilization of the lipophilic QT and MG up to 2.8 and 1.7 mg/mL (log P values 1.82 and 0.53, respectively) [20,46,47]. It is to be underlined that the loading of QT in POL enabled dramatic improvement of the polyphenol solubility with respect to water (being 0.00215 mg/mL at 25 °C) [48]. However, in all cases, to compare the different gels, the final drug concentration employed for further studies was 0.5 mg/mL (Table 1). The resulting gels appeared homogeneously yellow or white colored, respectively, in the case of QT- or MG-loaded semisolid forms. The HPLC chromatograms referring to the evaluation of QT and MG content in the different gels were free of secondary peaks ascribable to degradation products, confirming that the production protocol allows preservation of polyphenol stability.

### 3.2. In Vitro Polyphenol Diffusion Study

The capability of the semisolid formulation to control drug diffusion was investigated by an in vitro system based on Franz cells associated with a PTFE membrane [35]. This synthetic membrane model was employed due to its suitability to compare the performance of different topical forms, rather than to mimic skin permeation [34,36]. Particularly, QT or MG diffusion from the different gels was compared to the ethanolic drug solutions. As reported in Figure 4, almost all semisolid forms were able to hamper drug diffusion with respect to the simple solution, as expected due to their higher viscosity. Apart from the solution of MG, displaying a biphasic diffusion curve, steady-state diffusion profiles were observed.

Notably, the performance of POG was very similar, both in the case of QT- or MG-loaded forms, as appreciable considering the diffusion coefficient (D), the amount of drug diffused after 7 h (Q_7_) and the reduction ratio with respect to drug solution (Table 3), suggesting that in this case the gel viscosity exerts a rate-limiting effect on diffusion, irrespective of drug solubility. Conversely, different behaviors were detected in the case of QT and MG loaded in POL and OG, suggesting a different affinity of the drugs with the gel supramolecular structure. Particularly, in the case of QT, the diffusion parameters, reported in Table 3, followed the order SOL-QT > POL-QT > POG-QT > OG-QT, with a 21-fold decrease in drug diffusion in the case of OG-QT with respect to the simple solution. On the other hand, in the case of MG, the diffusion order was SOL-MG > OG-MG > POL-MG > POG-MG. Different considerations should be made regarding POL and OG. Indeed, in the case of POL-MG, D and Q_6_ values were almost double with respect to POL-QT, indicating that MG was less retained by POL with respect to QT. Notably, in the case of OG loaded with QT or MG, the performances of diffusion were dramatically different. Indeed, D and Q_7_ values for OG-QT were 10-fold lower with respect to OG-MG. These differences should be ascribed to the different solubilities of polyphenols and thus to their different extents of interaction within the supramolecular structure of OG [49].

### 3.3. Polyphenol Stability

In order to investigate the potential of semisolid forms to control QT and MG stability with respect to the simple ethanolic solutions, the drug content was analyzed in formulations stored at 25 °C for 4 months. In Figure 5a, the percentage amount of residual drug with respect to initial drug content in the different formulations is compared.

In general, MG was more prone to degradation with respect to QT, as indicated by residual drug content in the case of the drugs in solution. Notably, POL-QT was able to improve QT stability with respect to SOL-QT, maintaining 84% of drug content after 120 days. POL and POG better control MG stability with respect to SOL-MG, in agreement with diffusion kinetics. Conversely, OG was not suitable to control the degradation of both polyphenols. Notably, the obtained chromatograms were free of degradation products of QT and MG

### 3.4. Antioxidant Activity

Since in vivo studies conducted in mice and in rabbits have demonstrated that HSV-1 can cause oxidative stress [50], antioxidant activity of QT and MG in the different semisolid forms was evaluated by a PCL test. The results are summarized in Figure 5b and Table 4. As clearly appreciable, in general, the antioxidant capacity of QT-loaded forms was higher with respect to MG, namely, SOL-QT was almost six-fold more efficacious than SOL-MG. In particular, the most intense antioxidant potential was found in the case of POL-QT, followed by POG-QT.

### 3.5. In Vitro Antiviral Activity

A plaque reduction assay was used to evaluate in vitro anti-HSV-1 activities of QT- and MG-loaded forms. Particularly, a virucidal assay protocol was employed, evaluating direct action of formulations on the virus. SOL-QT, OG-QT, POL-QT and POG-QT were compared to the analogue MG-containing forms, using the same drug concentration (50 μg/mL). The formulations were in direct contact with the HSV-1 KOS strain at 35 °C before infection of the cell monolayer. The percent plaque reduction after 1 and 6 h of incubation was calculated (Table 4 and Figure 6).

After 1 h of contact with the virus and subsequent titration on the cell monolayer, SOL-QT exerted the most intense antiviral activity, 2.2-fold higher with respect to SOL-MG, and roughly double QT-loaded semisolid forms. Remarkably, prolonging the direct contact of the formulations with the virus to 6 h, the viral reduction of SOL-QT slightly decreased (13%), whilst an increase was achieved in the case of POL-QT and POG-QT, suggesting their capability to extend the virucidal activity. The prolonged effect on viral reduction was particularly appreciable in the case of POL-MG. Conversely, OG-QT displayed a different trend, maintaining the same virucidal activity after 6 h of incubation, as previously found in the case of OG-MG, that did not show any detectable effect [35]. Notably, the viral reduction obtained by unloaded gels was negligible, being below 20%, both at 1 and 6 h time points, suggesting that the gel excipients exhibit irrelevant anti-HSV-1 activity (Appendix A).

## 4. Discussion

It is well known that dermal delivery is strongly affected by the dosage form. Indeed, the choice of a delivery systems for topical application of poorly permeable drugs is crucial [51,52]. Notably, the selection of components represents a fundamental step in the case of the application on skin or lip lesions, such as those caused by HSV-1 [22]. Particularly, on one hand, the lipophilic character of polyphenols such as QT and MG requires a lipid phase in order to enhance their solubility and, on the other, aqueous formulations are easier to apply and more appreciated by the patients. In the present investigation, different specialized topical forms were considered in order to achieve these special requirements. All formulations were suitable for QT and MG solubilization and possessed an appropriate viscosity for topical application [35]. Notably, both POG and POL have thermosensitive behavior, with T_sol–gel_ 16.4 and 20.8 °C, respectively, but nonetheless, supposing that the handling occurs at about 20 °C, POL appears more indicated for handling and application on HSV-1 lesions. Notwithstanding the capability to control the in vitro drug diffusion exploited with all semisolid forms, some considerations should be made. In the case of POG, characterized by the highest viscosity, QT and MG diffusion parameters were almost superposable, indicating that the gel viscosity restrained the drug diffusion. Conversely, in the case of the less viscous POL and OG, the diffusion profiles of QT and MG were quite different. Indeed, in the case of POL-MG, D and Q_7_ values were almost double with respect to POL-QT, indicating that MG was less retained by POL with respect to QT. In this regard, it is likely that the higher MG polarity with respect to QT allows the molecule to diffuse faster through the gels towards the receiving phase constituted of ethanol 50% (*v*/*v*) and, moreover, a different affinity of the polyphenols for the gel supramolecular structure can be supposed [52]. Concerning this latest hypothesis, QT could be more firmly associated with the cubic ordered packing of POL with respect to MG, resulting in a slower diffusion. A peculiar behavior was found in the case of OG loaded with either QT or MG. As a general rule, the higher the vehicle viscosity, the slower the drug diffusion. Nonetheless, despite the gel’s lowest viscosity, QT diffusion from OG-QT was the slowest among all the semisolid forms. Furthermore, given the same gel, QT diffusion was 13-fold slower than MG, suggesting that, due to its higher lipophilicity, QT could be better retained within the entanglement of the tubular micellar structure formed by PC/IPP, while MG, being less retained, diffused more rapidly. Notably, it should be considered that the diffusion experiments were performed at 35 °C, a temperature suitable to mimic the skin condition, but very critical for OG rheological behavior, passing from a gel to liquid state (Figure 1a). Polyphenol stability was better controlled by POL, with the highest amount of drug retained in the case of POL-QT, confirming the firm association of QT with the cubic ordered structure of POL. This hypothesis was in agreement with antioxidant capacity, showing, in general, higher values in the case of QT-loaded forms with respect to MG-containing ones, with the highest antioxidant capacity in the case of POL-QT. Regarding the biological activity against HSV-1, a previous study demonstrated that neither the cell pretreatment with POL-MG, POG-MG, OG-MG and SOL-MG before viral infection, nor the addition of formulations during the adsorption phase, reduced the plaque formation [35]. For this reason, in the present investigation, the formulations were put in direct contact with HSV-1 KOS strains. The viral reduction results were quite interesting, especially after 6 h of contact, suggesting a possible action of QT and MG on the viral envelope, hampering the adsorption and penetration capacity of HSV-1 [18]. Notably, the capability of POL-QT to prolong the drug virucidal activity with respect to SOL-QT suggests that the loading of QT within the entangled semisolid form could prolong its therapeutic activity. On the other hand, the scarce virucidal effect exploited by OG-QT and OG-MG could be related to both the OG liquid state at 35 °C and to its oleaginous character that, in contact with the aqueous medium, led to phase separation, hindering the contact of the HSV-1 KOS strain with the loaded drugs [35].

## 5. Conclusions

The present study suggested the suitability of POL and POG as topical vehicles for polyphenols in the possible treatment or adjuvant therapy of HSV-1 infection. Both QT and MG were successfully loaded in P407- and PC-based gels, with a better ability of POL to solubilize QT and MG with respect to OG and POG. QT stability and antioxidant capacity were particularly improved by POL-QT, also showing a prolonged effect on viral reduction. It is likely that other polyphenols, such as resveratrol and curcumin, could be analogously employed in such gels for HSV-1 treatment but, nevertheless, each molecule needs a specific study. Indeed, despite similar physico-chemical characteristics, the mutual interactions of polyphenol with the supramolecular systems formed by PC and P407 could influence many factors, such as drug solubility and stability as well as antioxidant activity. Thus, we believe that each polyphenol requires a preformulative study. Moreover, further studies will enable studying the polyphenol permeability through 3D models of reconstituted human skin, as well as better investigation of the antiviral activity mechanism of polyphenol-loaded semisolid forms.

## Figures and Tables

**Figure 1 nanomaterials-12-00227-f001:**
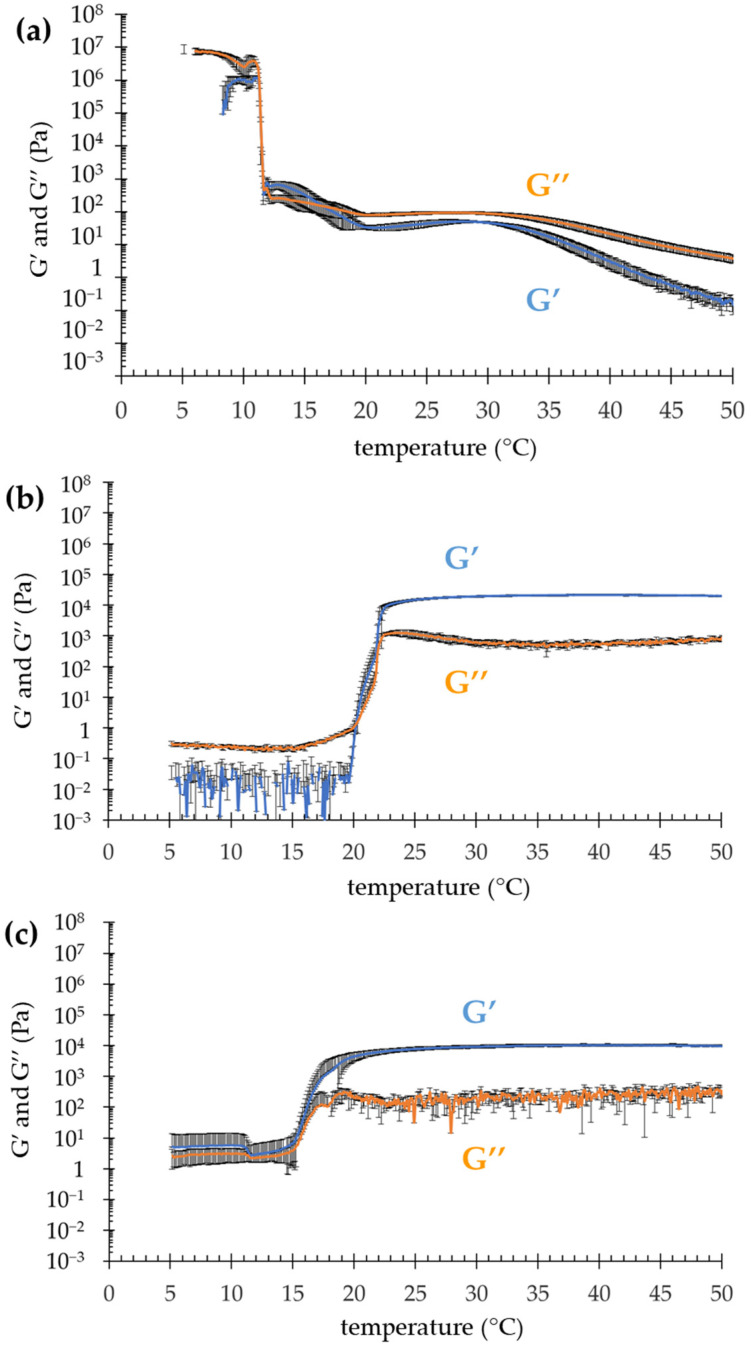
Evolution of elastic (G′) and viscous (G″) moduli as a function of temperature for OG (**a**), POL (**b**) and POG (**c**).

**Figure 2 nanomaterials-12-00227-f002:**
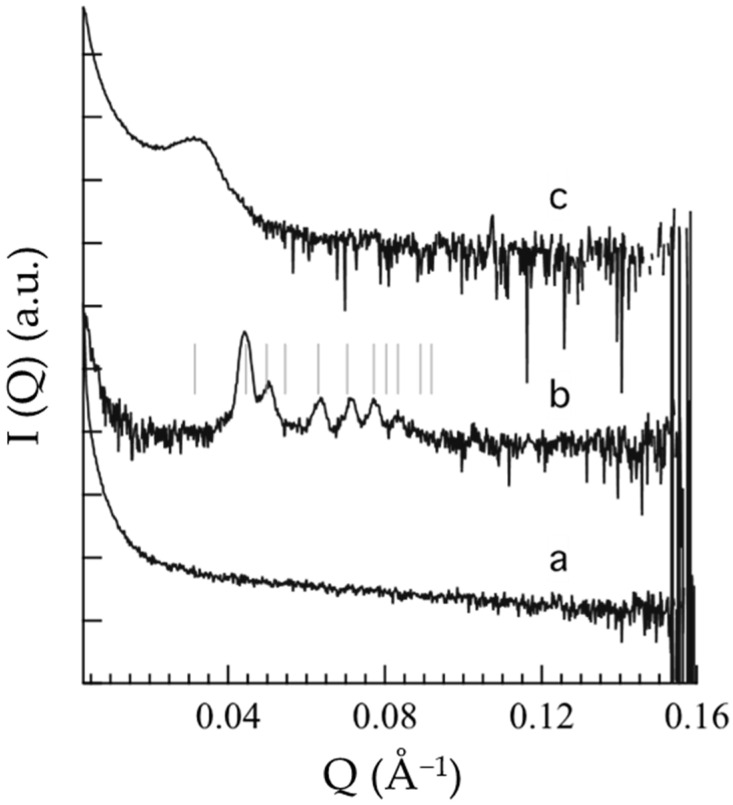
Small-angle X-ray scattering profiles of OG (**a**), POL (**b**) and POG (**c**). The thin vertical lines superposed to the b SAXS profile indicate the position of the Bragg peaks permitted for the Pm3n symmetry; note that Bragg peaks could be absent for other reasons than symmetry [41].

**Figure 3 nanomaterials-12-00227-f003:**
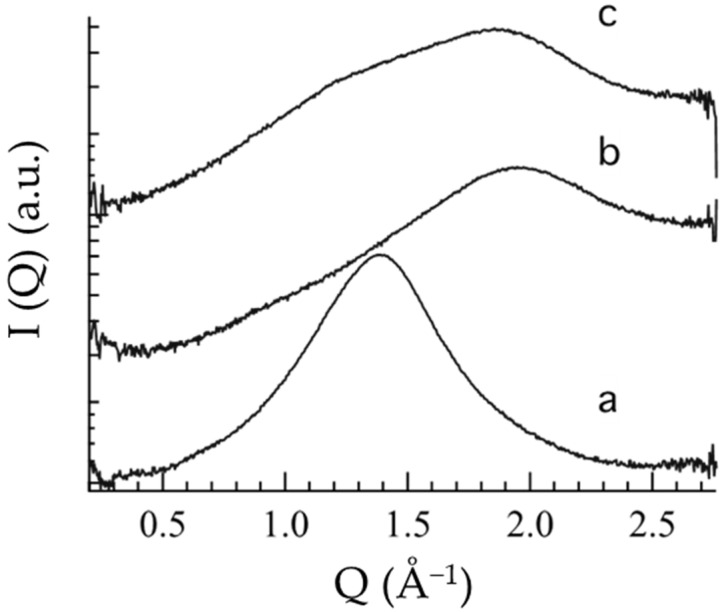
Wide-angle X-ray scattering profiles of OG (**a**), POL (**b**) and POG (**c**).

**Figure 4 nanomaterials-12-00227-f004:**
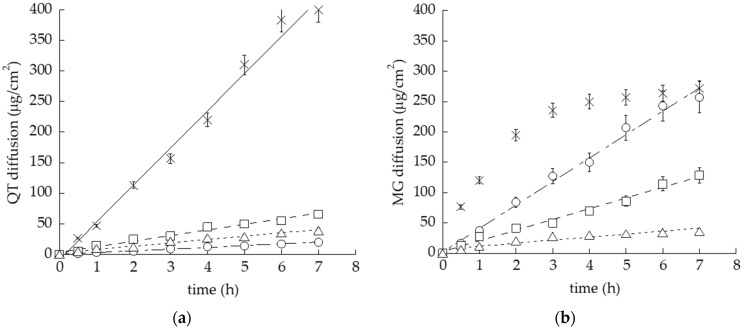
In vitro diffusion kinetics of QT (**a**) and MG (**b**) loaded in OG (circles), POL (squares), POG (triangles) or solubilized in ethanol/water, 30:70 *v/v* (crosses), as determined by Franz cells associated with PTFE membranes.

**Figure 5 nanomaterials-12-00227-f005:**
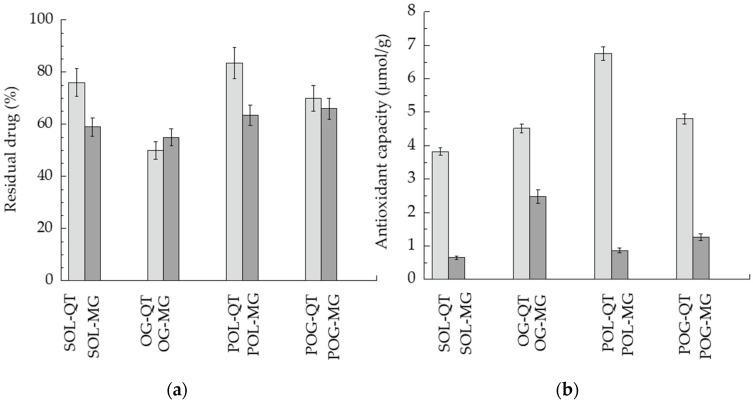
(**a**) QT (light gray) and MG (gray) content in formulations stored for 120 days at 25 °C, expressed as percentage with respect to the initial drug content. (**b**) Antioxidant capacity of QT (light gray) and MG (gray), as detected by ACL test, referring to μmol TE/g of sample.

**Figure 6 nanomaterials-12-00227-f006:**
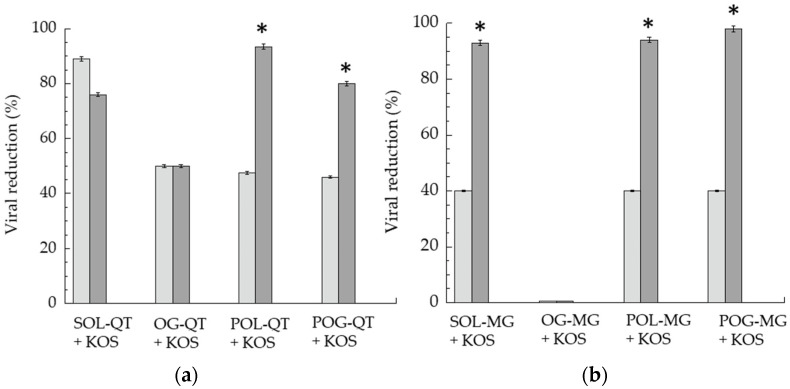
Virucidal activity of QT-containing (**a**) and MG-containing (**b**) formulations, expressed as percentage of viral reduction measured after 1 (light gray) and 6 (gray) h of contact between formulations and KOS strain at 35 °C. Data represent the mean of 3 independent experiments, * *p* values < 0.05.

**Table 1 nanomaterials-12-00227-t001:** Composition of semisolid vehicles employed for this study.

	Vehicle Composition (% *w*/*w*)								
Component	OG	POL	POG	OG-QT	POL-QT	POG-QT	OG-MG	POL-MG	POG-MG
PC ^1^	15.60	-	4.68	15.60	-	4.68	15.60	-	4.68
P-407 ^2^	-	20.0	14.00	-	20.00	14.00	-	20.0	14.00
IPP ^3^	82.51	-	24.82	82.46	-	24.82	82.46	-	24.82
water	1.44	80.0	56.50	1.44	79.95	56.45	1.44	79.95	56.45
QT ^4^	-	-	-	0.05	0.05	0.05	-	-	-
MG ^5^	-	-	-	-	-	-	0.05	0.05	0.05

^1^: soy phosphatidylcholine; ^2^: poloxamer 407; ^3^: isopropylpalmitate; ^4^: quercetin; ^5^: mangiferin.

**Table 2 nanomaterials-12-00227-t002:** Rheological and diffraction parameters of the indicated forms.

Vehicle	T_sol–gel_ ^1^ (°C)	Structure ^2^	SAXS Repeat Distance (Å) ^2^	WAXS Repeat Distance (Å) ^2^
OG	11.5 ± 0.1 16.4 ± 0.7	Disordered micellar	-	4.59
POL	20.8 ± 0.7	Cubic (space group Q223)	282.08	3.29
POG	16.4 ± 2.1	Ordered micellar	196.34	3.34–4.83

^1^: sol–gel transition temperature; ^2^: determined at 25 °C.

**Table 3 nanomaterials-12-00227-t003:** Diffusion parameters of the indicated formulations.

Vehicles	F ^1^ ± s.d. (μg/cm^2^/h)	QT (mg/mL)	MG (mg/mL)	D ^2^ ± s.d. (cm/h) × 10^−3^	Q_7_ ^3^ ± s.d. (μg/cm^2^)	Reduction Ratio ^4^
OG-QT	2.84 ± 1.1	0.5	-	5.68 ± 2.2	20 ± 4	21.5
POL-QT	9.16 ± 2.7	0.5	-	18.32 ± 5.4	66 ± 11	6.6
POG-QT	5.42 ± 4.3	0.5	-	10.84 ± 8.6	39 ± 17	12.2
SOL-QT	61.06 ± 2.8	0.5	-	122.13 ± 5.6	403 ± 32	-
OG-MG	38.69 ± 6.8	-	0.5	77.38 ± 13.6	257.14 ± 38	1.96
POL-MG	17.64 ± 2.4	-	0.5	35.28 ± 4.8	128.57 ± 10	4.3
POG-MG	5.02 ± 1.1	-	0.5	10.04 ± 2.2	36.43 ± 4	15.11
SOL-MG	75.86 ± 12	-	0.5	151.72 ± 24	271.43 ± 23	-

^1^: Flux; ^2^: diffusion coefficient; ^3^: drug diffused after 7 h; ^4^: reduction in diffusion with respect to SOL-MG or SOL-QT; data are the mean of 6 independent Franz cell experiments.

**Table 4 nanomaterials-12-00227-t004:** Antioxidant and virucidal parameters of the indicated formulations.

Vehicles	ACL (μmol TE/g)	Plaque Reduction (%)	
		1 h	6 h
OG-QT	4.52 ± 0.34	50.0 ± 1.0	50.0 ± 1.0
POL-QT	6.76 ± 0.14	47.5 ± 1.0	93.5 ± 0.75
POG-QT	4.80 ± 0.23	46.0 ± 0.75	80.0 ± 0.75
SOL- QT	3.82 ± 0.05	89.0 ± 1.0	76.0 ± 1.0
OG-MG	2.48 ± 0.02	-	-
POL-MG	0.87 ± 0.02	40.0 ± 1.7	94.0 ± 0.74
POG-MG	1.26 ± 0.06	40.0 ± 1.0	98.0 ± 0.62
SOL-MG	0.65 ± 0.04	40.0 ± 1.7	93.0 ± 0.74

QT and MG: 0.5 mg/mL.

## Data Availability

Not applicable.

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
