# Peer review of "Natural Polyphenol-Containing Gels against HSV-1 Infection: A Comparative Study"

_nanomaterials, 2022, doi:10.3390/nano12020227_

Round 1

Reviewer 1 Report

Dear Authors,

Thank you for the interesting manuscript.

I only recommend to fix some points in the manuscript:

  1. The Conclusions section has to be extended for the readers who try to briefly understand the main results of the study.
  2. Figure captions for Figures 2, 3 and 5 are separated from the following Figures.
  3. Lines 174, 178, 179, 181 and further throughout the entire text. "ml" and "μl" has to be changed to "mL" and "μL", accordingly, as it usually used in the technical journals.
  4. Line 432. Table 4 heading separator has to be changed to dot ("Table 4. Antioxidant...").
  5. Equation (2) line has to be finished with comma.
  6. Lines 218, 221, 235, 249, 253 and throughout the entire text. "x" has to be changed to multiplication symbol "×".
  7. Lines 180 and 230. "rpm" has to be changed to "min–1".
  8. Line 204. "mWatt/cm2" has to be changed to "mW/cm2".
  9. Line 241. Term "Sigma" has to be specified (do the Authors mean "Sigma-Aldrich"?").
  10. Table 3. "(cm/h)*10-3" has to be changed to "(cm/h)×103".

Author Response

Response to Reviewer 1 Comments

I only recommend to fix some points in the manuscript:

We thank the reviewer for his comment and suggestions.

Point 1. The Conclusions section has to be extended for the readers who try to briefly understand the main results of the study.

Response 1. We improved the conclusion section, accordingly (lines 518-521).

 Point 2. Figure captions for Figures 2, 3 and 5 are separated from the following Figures.

Response 2. We are sorry, probably there was a formatting drawback. In the original and in the revised version of the manuscript the figure captions are in the right place.

Point 3. Lines 174, 178, 179, 181 and further throughout the entire text. "ml" and "μl" has to be changed to "mL" and "μL", accordingly, as it usually used in the technical journals.

Response 3. Throughout the manuscript "ml" and "μl" were changed to "mL" and "μL", accordingly.

 Point 4. Line 432. Table 4 heading separator has to be changed to dot ("Table 4. Antioxidant...").

Response 4. In Table 4 heading separator was changed to dot.

Point 5. Equation (2) line has to be finished with comma.

Response 5. Equation (2) line has been finished with comma.

Point 6. Lines 218, 221, 235, 249, 253 and throughout the entire text. "x" has to be changed to multiplication symbol "×".

Response 6. Throughout the entire text. "x" has been changed to multiplication symbol "×".

Point 7. Lines 180 and 230. "rpm" has to be changed to "min–1".

Response 7. "rpm" has to be changed to "min–1", accordingly.

 Point 8. Line 204. "mWatt/cm2" has to be changed to "mW/cm2".

Response 8. mWatt/cm2" has been changed to "mW/cm2.

Point 9. Line 241. Term "Sigma" has to be specified (do the Authors mean "Sigma-Aldrich"?").

Response 9. Yes, we mean Sigma-Aldrich, the term was changed accordingly.

Point 10. Table 3. "(cm/h)*10-3" has to be changed to "(cm/h)×10–3".

Response 10. (cm/h)*10-3" has been changed to "(cm/h)×10–3", accordingly.

Reviewer 2 Report

The manuscript is well written and compared the organogels, POL and poloxamer organogel as delivery systems for quercetin and mangiferin. The authors studied the rheology of the gels, while the inner structure was evaluated by x-ray diffraction. Also, vitro the effect of drug loading in semisolid forms studied in vitro. The biological efficacy of quercetin or mangiferin-loaded forms was evaluated.

The results are very interesting.

Accordingly, the manuscript can be accepted.

Author Response

Response to Reviewer 2 Comments

The manuscript is well written and compared the organogels, POL and poloxamer organogel as delivery systems for quercetin and mangiferin. The authors studied the rheology of the gels, while the inner structure was evaluated by x-ray diffraction. Also, vitro the effect of drug loading in semisolid forms studied in vitro. The biological efficacy of quercetin or mangiferin-loaded forms was evaluated.

The results are very interesting.

Accordingly, the manuscript can be accepted.

We thank the reviewer for his kind comment.

Reviewer 3 Report

This study proposes a new approach to combine phospholipids and surfactants using plant-derived polyphenols as a new technology against HIV-1 virus. The use of polyphenols as a new technology beyond the drug acyclovir, which is widely used as a drug, is significant, and the paper provides a new perspective for the reader. The paper is well written, but the reviewer suggests the following comments to help the reader better understand the importance of this paper.

Comment 1:

In this study, quercetin and mangiferin are used as models of plant-derived polyphenols, but reviewer think an explanation in the text of why these two polyphenols were selected from among the many polyphenols with anti-HIV activity is essential. Were these polyphenols chosen by the authors because quercetin and mangiferin are the most powerful of the polyphenols? For example, if other polyphenols such as curcumin or resveratrol, which have similar physical properties (fat-soluble), are used instead of quercetin and mangiferin, can the anti-HIV-1 viral activity be confirmed with this gel technology? It would be good to add some information and discussion, especially since there are many researchers on curcumin and HIV-1.

Examples of relevant literature are reported below:

Curcumin inhibits HIV-1 by promoting Tat protein degradation

https://www.nature.com/articles/srep27539

doi: 10.1038/srep27539

Potent Inhibition of HIV-1 Replication in Resting CD4 T Cells by Resveratrol and Pterostilbene

https://www.ncbi.nlm.nih.gov/pmc/articles/PMC5571302/

doi: 10.1128/AAC.00408-17

Comment 2:

In the introduction, the authors explain why Poloxamer 407 was chosen in this study. The attractive property of Poloxamer 407 makes it worthy to be used in this study. On the other hand, many reports are concerned about its safety of nonionic surfactant, compared to phospholipids. In the text, the safety of surfactants seems to have been insufficiently explained to the readers. Last year, an extensive review of nonionic surfactants for food and pharmaceutical applications was reported. This review supports the low toxicity of the surfactants that the authors used in the experiment.

A Critical Review of the Use of Surfactant-Coated Nanoparticles in Nanomedicine and Food Nanotechnology

https://www.dovepress.com/a-critical-review-of-the-use-of-surfactant-coated-nanoparticles-in-nan-peer-reviewed-fulltext-article-IJN

doi: 10.2147/IJN.S298606

In the introduction, reviewer recommend authors to cite this article and emphasize that the surfactant used in this experiment are appropriate (low toxicity).

Comment 3:

The reviewer is very concerned about the effect of temperature on experimental procedure. In the methods section, 2.2. Preparation of semisolid forms, the authors mixed quercetin and mangiferin at 25°C for the preparation of lecithin organogel. When preparing Poloxamer gel, the authors mixed quercetin and mangiferin at 4°C. In other words, these different gels are prepared under different temperatures. The authors stated that the content of quercetin and mangiferin in these gels is 0.05 % (w/w) in Table 1. Do organogels have a better ability to solubilize quercetin and mangiferin than Poloxamer gel?

On the other hand, if the following previous report on the effect of temperature on the solubility of quercetin is confirmed, the solubility of quercetin in water at 25°C seems to be about 0.000215 % (w/w).

Solubility and solution thermodynamic properties of quercetin and quercetin dihydrate in subcritical water

https://www.sciencedirect.com/science/article/pii/S0260877410001822

doi: 10.1016/j.jfoodeng.2010.04.001

In light of this previous report, the reviewer is concerned that if the temperature drops during the experimental procedures, the crystals of quercetin and mangiferin may no longer be in a soluble state and may precipitate in the system. Reviewer suggest that temperature information be included in all of the experimental procedures performed by the authors. Are there significant effects of temperature when doing the lecithin organogel and Poloxamer gel preparation with quercetin and mangiferin in this study? Reviewer suggest that this temperature effect should also be explained in the discussion section.

Comment 4:

In the section 2.7. HPLC Analysis, the author describes the information on the analytical method for quercetin and mangiferin by HPLC, but the sample preparation procedures is needed.  Did authors dilute the sample standard or gel directly into the composition of the mobile phase and analyze that? Did you filter the sample before analysis? Furthermore, reviewer suggest that authors prepare new Figure of HPLC chromatogram of quercetin and mangiferin extracted from the gel. Showing a chromatogram free of degradation products of quercetin and mangiferin would emphasize the usefulness of this gel preparation method.

Comment 5:

The nonionic surfactant COL-1492 was found to be useful against HIV-1 in vitro, but a famous previous report in 2002 stated that human studies were conducted for anti-HIV-1 activity and that no in vivo efficacy was confirmed.

Effectiveness of COL-1492, a nonoxynol-9 vaginal gel, on HIV-1 transmission in female sex workers: a randomised controlled trial

https://pubmed.ncbi.nlm.nih.gov/12383665/

doi: 10.1016/s0140-6736(02)11079-8

Given that there are such reports showing activity with only such surfactants. In present report, authors only compared the results between the groups containing quercetin and mangiferin, and did not examine the virucidal activity in the group without them. Is there any possibility that a dispersion of the surfactant (Poloxamer 407) or even a Poloxamer gel-only group (without polyphenols) would exhibit anti-HIV-1 activity in vitro? Reviewer think that data or discussion on these is needed.

Author Response

Response to Reviewer 3 Comments

This study proposes a new approach to combine phospholipids and surfactants using plant-derived polyphenols as a new technology against HIV-1 virus. The use of polyphenols as a new technology beyond the drug acyclovir, which is widely used as a drug, is significant, and the paper provides a new perspective for the reader. The paper is well written, but the reviewer suggests the following comments to help the reader better understand the importance of this paper.

We thank the reviewer for his comment and suggestions.

Point 1 In this study, quercetin and mangiferin are used as models of plant-derived polyphenols, but reviewer think an explanation in the text of why these two polyphenols were selected from among the many polyphenols with anti-HIV activity is essential. Were these polyphenols chosen by the authors because quercetin and mangiferin are the most powerful of the polyphenols? For example, if other polyphenols such as curcumin or resveratrol, which have similar physical properties (fat-soluble), are used instead of quercetin and mangiferin, can the anti-HIV-1 viral activity be confirmed with this gel technology? It would be good to add some information and discussion, especially since there are many researchers on curcumin and HIV-1.

Examples of relevant literature are reported below:

Curcumin inhibits HIV-1 by promoting Tat protein degradation

https://www.nature.com/articles/srep27539

doi: 10.1038/srep27539

Potent Inhibition of HIV-1 Replication in Resting CD4 T Cells by Resveratrol and Pterostilbene

https://www.ncbi.nlm.nih.gov/pmc/articles/PMC5571302/

doi: 10.1128/AAC.00408-17

Response 1. We thank the reviewer for these comments. Nevertheless we would like to underline that the object of the manuscript is a new technology against HSV-1 virus instead of HIV-1 virus. As you know HSV-1 and HIV-1 viruses have completely different pathogenetic mechanisms of infection, transmission and symptomatology. In a previous paper we decided to study mangiferin, being a relatively new, little exploited antioxidant, afterwards, we compared it to quercetin that is well-known for its strong antioxidant potential, as well as for many other therapeutic activities, such as the antiviral one.  Anyway, we agree, it is likely that other polyphenols such as resveratrol and curcumin could be analogously employed in such gels for HSV-1 treatment. Nevertheless, each molecule would need a specific study since, despite similar physico-chemical characteristics, P407 could influence many factors, such as drug solubility, stability as well as antioxidant activity. Thus, we believe that a preformulative study should be required for each polyphenol. A brief paragraph has been added in the conclusion section at this regard (lines 522-527).

Point 2: In the introduction, the authors explain why Poloxamer 407 was chosen in this study. The attractive property of Poloxamer 407 makes it worthy to be used in this study. On the other hand, many reports are concerned about its safety of nonionic surfactant, compared to phospholipids. In the text, the safety of surfactants seems to have been insufficiently explained to the readers. Last year, an extensive review of nonionic surfactants for food and pharmaceutical applications was reported. This review supports the low toxicity of the surfactants that the authors used in the experiment. 

A Critical Review of the Use of Surfactant-Coated Nanoparticles in Nanomedicine and Food Nanotechnology

https://www.dovepress.com/a-critical-review-of-the-use-of-surfactant-coated-nanoparticles-in-nan-peer-reviewed-fulltext-article-IJN

doi: 10.2147/IJN.S298606 

In the introduction, reviewer recommend authors to cite this article and emphasize that the surfactant used in this experiment are appropriate (low toxicity).

 Response 2. We thank the reviewer for this suggestion. Accordingly, at lines 68-70 a phrase has been added and the review has been cited. 

Point 3:The reviewer is very concerned about the effect of temperature on experimental procedure. In the methods section, 2.2. Preparation of semisolid forms, the authors mixed quercetin and mangiferin at 25°C for the preparation of lecithin organogel. When preparing Poloxamer gel, the authors mixed quercetin and mangiferin at 4°C. In other words, these different gels are prepared under different temperatures. The authors stated that the content of quercetin and mangiferin in these gels is 0.05 % (w/w) in Table 1. Do organogels have a better ability to solubilize quercetin and mangiferin than Poloxamer gel?

On the other hand, if the following previous report on the effect of temperature on the solubility of quercetin is confirmed, the solubility of quercetin in water at 25°C seems to be about 0.000215 % (w/w).

 Solubility and solution thermodynamic properties of quercetin and quercetin dihydrate in subcritical water

https://www.sciencedirect.com/science/article/pii/S0260877410001822

doi: 10.1016/j.jfoodeng.2010.04.001

In light of this previous report, the reviewer is concerned that if the temperature drops during the experimental procedures, the crystals of quercetin and mangiferin may no longer be in a soluble state and may precipitate in the system. Reviewer suggest that temperature information be included in all of the experimental procedures performed by the authors. Are there significant effects of temperature when doing the lecithin organogel and Poloxamer gel preparation with quercetin and mangiferin in this study? Reviewer suggest that this temperature effect should also be explained in the discussion section.

Response 3. With regard to the preparation of drug-loaded gels, we would like to underline that quercetin and mangiferin were not mixed at 4 °C! Actually, in the original manuscript the temperature was not specified, while in the revised version, at line 124, it has been specified that the drugs were added and mixed under stirring at 25 °C. The unloaded gel was prepared at 4° C, while quercetin and mangiferin were solubilized in the preformed gel at room temperature. In addition as specified in the original manuscript “… the micellar structure formed by the block co-polymer allowed to solubilize the lipophilic QT and MG up to 2.8 and 1.7 mg/mL”. Thus if we consider the result of the paper suggested by the reviewer, the loading of quercetin in poloxamer gel enabled to 1000-fold improve its solubility with respect to water! This solubility power was achieved due to the block co-polymer P407. Thus poloxamer gel has a better ability to solubilize quercetin and mangiferin with respect to organogel. Indeed, it is well known that surfactants at concentration above their critical micelle  concentration (c.m.c.) form micelles that allow to improve the solubility of lipophilic drugs in hydrophilic solvent, such as water. The phrase “It is to be underlined that the loading of QT in POL enabled to dramatically improve the polyphenol solubility with respect to water (being 0.00215 mg/mL at 25 °C)” has been added at lines 366-368. In addition, in the revised version we specified that “To prepare OG-QT and OG-MG, the drugs were solubilized in the lipophilic phase constituted of PC/IPP before the addition of water, reaching 0.8 and 0.7 mg/mL for QT and MG respectively” (lines 361-362). Thus the solubilization of polyphenols was better achieved by poloxamer gel, rather than by organogel. Moreover, the reviewer referred to 0.05 % (w/w) in Table 1: this is the drug content in the vehicle composition, that is different from the maximum drug solubility within the vehicles. In order to better specified this point, the phrase “However, in all cases, to compare the different gels, the final drug concentration em-ployed for further studies was 0.5 mg/mL.” has been added at lines 368-370.

With regard to the possibility of crystals of quercetin and mangiferin precipitating in the system, we would like to underline that the transparency of both poloxamer gel and organogel allowed to detect the total absence of drug crystals below the above reported drug concentrations. In the conclusion section, the better ability of Poloxamer gel to solubilize quercetin and mangiferin with respect to organogels has been mentioned. The temperature information has been included in all the experimental procedures performed and evidenced in yellow. In the results and in the discussion section the temperature effect on polyphenol solubilization has not been described, since quercetin and mangiferin were solubilized in the gels at the same temperature (25 °C), whilst the gel thermoresponsive behavior was already thoroughly described in the original manuscript.

Point 4: In the section 2.7. HPLC Analysis, the author describes the information on the analytical method for quercetin and mangiferin by HPLC, but the sample preparation procedures is needed.  Did authors dilute the sample standard or gel directly into the composition of the mobile phase and analyze that? Did you filter the sample before analysis? Furthermore, reviewer suggest that authors prepare new Figure of HPLC chromatogram of quercetin and mangiferin extracted from the gel. Showing a chromatogram free of degradation products of quercetin and mangiferin would emphasize the usefulness of this gel preparation method.

Response 4. As reported in the original manuscript, at lines 160-164, section 2.5. Polyphenol content of semisolid forms, “QT and MG entrapment capacity (EC) was evaluated after 1 and 120 days from semi-solid form preparation, as previously reported [34]. Briefly, the forms were diluted with ethanol in a 1:10 w/w ratio, stirred for 30 min, and filtered by nylon syringe filters (0.22 μm pores).” In addition the following phrase was added to the 2.7 Section: “Before injection, the samples were appropriately diluted in the mobile phases” (line 197).

In order to accomplish the reviewer suggestion, we tried to prepare a Figure with chromatograms of quercetin and mangiferin, but we believe that the quality of the Figure is too poor. Unfortunately, we had to take images from the screen of PC, since the operative system we employ is obsolete and does not allow to copy chromatogram files. Therefore, to follow the reviewer’s insightful suggestion, the following phrase was added: “The HPLC chromatograms referring to the evaluation of QT and MG content in the different gels were free of secondary peaks ascribable to degradation products, confirming that the production protocol allowed to preserve polyphenol stability. " (lines 371-374).

Point 5: The nonionic surfactant COL-1492 was found to be useful against HIV-1 in vitro, but a famous previous report in 2002 stated that human studies were conducted for anti-HIV-1 activity and that no in vivo efficacy was confirmed. Effectiveness of COL-1492, a nonoxynol-9 vaginal gel, on HIV-1 transmission in female sex workers: a randomised controlled trial https://pubmed.ncbi.nlm.nih.gov/12383665/ doi: 10.1016/s0140-6736(02)11079-8.

Given that there are such reports showing activity with only such surfactants. In present report, authors only compared the results between the groups containing quercetin and mangiferin, and did not examine the virucidal activity in the group without them. Is there any possibility that a dispersion of the surfactant (Poloxamer 407) or even a Poloxamer gel-only group (without polyphenols) would exhibit anti-HIV-1 activity in vitro? Reviewer think that data or discussion on these is needed.

Response 5: As above reported, we would like to underline that the reviewer refers to HIV-1, while the object of the manuscript is a new technology against HSV-1 virus rather than HIV-1 virus.

Anyway, to address to the interesting point raised by the reviewer, Figure S1 has been added as Supplementary material. “Notably, the viral reduction obtained by unloaded gels was negligible, being below 20 %, both at 1 and 6 h time points, suggesting that the gel excipients exhibit irrelevant anti-HSV-1 activity (Fig. S1).” This phrase was inserted at lines 460-462.

Round 2

Reviewer 1 Report

Dear Authors,

Thank you for addressing my comments. I believe the revised manuscript can be accepted for publication.